# Were ancient foxes far more carnivorous than recent ones?—Carnassial morphological evidence

Elwira Szuma[ID][1]*, Mietje Germonpré[ID][2]

**1** Mammal Research Institute Polish Academy of Sciences, Białowieża, Poland, **2** Operational Direction "Earth and History of Life", Royal Belgian Institute of Natural Sciences, Brussels, Belgium

* eszuma@ibs.bialowieza.pl

## Abstract

Crown shape variation of the first lower molar in the arctic (*Vulpes lagopus*) and red foxes (*Vulpes vulpes*) was analyzed using five groups of morphotypes. Carnassial morphologies were compared between the species and between spatially and temporally distant populations: one Late Pleistocene (n = 45) and seven modern populations of the arctic fox (n = 259), and one Late Pleistocene (n = 35) and eight modern populations of the red fox (n = 606). The dentition of Holocene red foxes had larger morphotype variability than that of arctic foxes. The lower carnassials of the red fox kept have some primitive characters (additional cusps and stylids, complex shape of transverse cristid), whereas the first lower molars of the arctic fox have undergone crown shape simplification, with the occlusal part of the tooth undergoing a more pronounced adaptation to a more carnivorous diet. From the Late Pleistocene of Belgium to the present days, the arctic fox's crown shape has been simplified and some primitive characters have disappeared. In the red fox chronological changes in the morphology of the lower carnassials were not clearly identified. The phyletic tree based on morphotype carnassial characteristics indicated the distinctiveness of both foxes: in the arctic fox line, the ancient population from Belgium and recent Greenland made separate branches, whereas in the red foxes the ancient population from Belgium was most similar to modern red foxes from Belgium and Italy.

## Introduction

The carnassials–the upper fourth premolars ($P^4$) and the opposite first lower molars ($M_1$) are specialized and very important in the dental systems of carnivorous mammals. This pair of teeth occupies a central position in the tooth-row, which gives them advantage by maximizing bite force and gape. The main function of the carnassials is slicing and grinding of food; thus, the shape of the occlusal crown surface is intimately associated with dietary preferences in carnivores.

Contemporary studies of the types of food to which teeth and skulls are adapted use both traditional and advanced approaches. Traditional analyses of tooth function are based on

**Data Availability Statement:** All relevant data are within the manuscript and its Supporting Information files.

**Funding:** YES: Data to the study (fossil materials, and resent foxes from Belgium, RBINS) were

collected with financial support of the SYNTHESYS BE-TAF project (2006); Recent materials used to comparative analyses originated from many different institutions and museums. The data were collected by financial support of the State Committee for Scientific Research (grant No. P04C 063 20)

**Competing interests:** The authors have declared that no competing interests exist.

measuring the size, size proportions, and angles of teeth [1, 2]. Recently, two-dimensional geometric morphometric [3–5] or three-dimensional modeling [6, 7] have often been used for analyzing crown complexity. The established approach relies on defining tooth shape into morphotype units [8–10]. This approach requires thorough knowledge of the dental morphology of the target species. Up till now morphotype analyses have been carried out during research of the dental characteristics of modern mammal species such as the red fox *Vulpes vulpes* [11–15]; arctic fox *Vulpes lagopus* [14–16]; corsac *Vulpes corsac* [14, 15]; raccoon dog *Nyctereutes procyonoides* [17]; European pine marten *Martes martes*, stone marten *Martes foina* and sable *Martes zibellina* [18, 19]; Japanese marten *Martes melampus*, American pine marten *Martes americana*, yellow-throated marten *Martes flavigula*, and fisher *Martes pennanti* [18]; badger *Meles meles* [20, 21]; brown bear *Ursus arctos* and arctic bear *Ursus maritimus* [22]; arctic ground squirrel *Citellus parryi* [23]; hares *Lepus* sp. [24–27]; and some vole species *Microtinae* [10, 28–32]. Such detailed analyses of the tooth morphologies of recent mammals are useful for studying fossil dental remains. Knowledge of tooth structure morphologies and evolutionary, ecological, behavioral and physiological constraints of the dental patterns of modern species enables us to picture the habitats and adaptations of extinct mammal species of past epochs.

Morphotype studies of recent materials are useful, especially as most identifications and classifications of extinct mammals are based on fossil remains such as molars and premolars, and pieces of mandibles with teeth [15, 33–36]. Morphotype fractions like another measure–orientation patch count (OPC) can be used to quantify variation in dental complexity for many groups of mammals [6, 12]. Variation in the morphological dental patterns of mammals can be analyzed for various levels of systematic and ecological units. In each mammal species, subspecies, and population, specific dental patterns are a result of the multilateral effects of different factors, such as evolutionary history, food habits, climate, intra and inter-specific competition, and population density. Some shifts in tooth structure complexity between mammal taxa, subspecies, and geographically or chronologically separated populations of the same species are direct phenotypic responses to changes in the natural environment [13, 37]. Analyses of tooth occlusal surfaces (the complicated patterns of their crests and cusps) in microchiroptera, insectivore, carnivore, primate, and marsupial species indicate that more conspicuous and stronger occlusal tooth crown structures are related to harder foods; whereas, lower cusps and more complicated enamel crest patterns are related to consuming much softer-objects [38, 39]. Over the last century, some shifts have been registered in the morphotype frequencies of the dental pattern of the red fox in Poland [37]. The observations reflect the red fox's broadening food niche and more optimal utilization of anthropogenic habitats. Subsequent studies on European red foxes have confirmed these observations. Over the last 50 years skull and body size in recent red foxes have grown in different populations [40, 41]. Both the temporal shift in dental morphology to a more generalist pattern and the increase in skull and body size of the red fox are a result of higher food availability and clear commensalism with man in the last decades of the last century.

Morphotype analysis of tooth shape in mammals can directly show dietary preferences and also reliably characterize their natural habitat. Morphological dental patterns constructed at the population, subspecies or species levels offer the opportunity to study phyletic or phylogeographic relationships and distances. Szuma [13], based on morphotype dental patterns of the red fox, showed variation between populations in carnivore specialization level and indicated the most probable place of diversification of the *V. vulpes* line: the area between Asia Minor and the Middle East. Morphotype characterization of the red fox throughout its whole native range uncovered the phylogeographic relations of the species. Analogical analysis was also carried out for the arctic fox across almost the whole of its Arctic range [16].

The aim of this research was to characterize the morphotype variation of the lower carnassials in the red and arctic foxes during the Late Pleistocene of Belgium. The morphotype patterns of the first lower molars in ancient samples were compared with morphological patterns found previously in extant populations of both species [13, 16]. Up till now there have been no studies on the phylogenetic relationships between Late Pleistocene arctic foxes in Europe. We expected to uncover reliable phyletic relationships between recent and fossil populations based on the crown shape of the lower carnassials. In our view the dental carnivorous specialization of the arctic and red foxes has weakened over the epochs following the Pleistocene. Over the last century the dietary opportunism of the European red fox has increased. Moreover, the dental pattern of Polish red foxes has been found to have become less carnivorous [37]. Based on our results, we discuss the evolutionary changes in the dental patterns of both foxes from the Pleistocene to present.

## Material and methods

In this study we used fossil material of the red (*V*. *vulpes* Linnaeus, 1758) and arctic foxes (*V*. *lagopus* Linnaeus, 1758) from the Late Pleistocene of Belgium and recent material of the two species from geographically distinct populations. The fossil sample of the arctic fox contained 45 lower carnassials from four Palaeolithic sites (Trou de Chaleux, Goyet, Trou des Nutons, Trou Magritte; S1 Table). The fossil material of the red fox consisted of 35 isolated first lower molars from four Upper Palaeolithic sites (Trou de Chaleux, Goyet, Trou du Frontal, Trou des Nutons) in Belgium (S2 Table). Some carnassials could be from one specimen (the same fox): left and right carnassials; we studied isolated teeth so we could not assess specimen number. We analyzed left and right teeth together, as all specimens were equally valuable for drawing good conclusions. Morphological dental characters generally indicated bilateral symmetry; however there were also many cases of unilateral expression of morphotype characters. Determining how the collective analysis of left and right teeth impacted the study outcomes is difficult. However, the number of ancient samples of the red arctic fox seems to have been large enough to minimize the effect of the method.

All these fossil sites are in the Ardennes Massif and contain Pleistocene fauna. Additionally, a sample of 61 skulls of recent red foxes from Belgium was from the collections of the Royal Belgian Institute of Natural Sciences (RBINS).

The Trou de Chaleux cave is situated on the right bank of the River Lesse, a tributary of the Meuse River [42]. Inside this cave a major bone bed with a tremendous number of skeletal remains was discovered by Dupont [43]. Several AMS (Accelerator Mass Spectrometry) dates are available with calibrated ages ranging from 15 840 cal BP to 14 130 cal BP (S3 Table). The calibrations were calculated using the Oxcal 3.4 program (https://c14.arch.ox.ac.uk/oxcal/OxCal.html). Arctic fox remains represent 11.5% of the number of identified specimens (NISP) assigned to the Pleistocene fauna, whereas red fox remains account for 1.9% [44].

The Goyet cave forms part of a large karstic system on the right bank of the Samson valley, a tributary of the Meuse River. In this cave numerous Pleistocene mammal bones and human remains, and large quantities of Middle and Upper Palaeolithic artefacts have been found [45–47]. The calibrated ages range from 44 990 cal BP to 14 460 cal BP for Horizon 1 and from 40 210 cal BP to 14 120 cal BP for Horizon 2 (S3 Table). Foxes constitute 3.8% of the NISP from Bed 1 [44, 48] and 3.7% of the NISP from Bed 2 ([44]; Germonpré unpublished data).

The Trou des Nutons cave is situated on the right bank of the Lesse River. The main bone bed yielded Magdalenian artefacts, and the Pleistocene mammal assemblage includes horse (*Equus* sp.) and reindeer (*Rangiffer tarandus*) bones [49]. Nevertheless, the AMS dates indicate that this assemblage consists of different components with calibrated ages ranging from 26 230

cal BP to 14 240 cal BP ([49, 50]; S3 Table). The bones of the arctic and red foxes constitute resp. 27.8% and 11.5% of the NISP of the Pleistocene assemblage (Germonpré unpublished data).

The Trou du Frontal cave is located on the right bank of the Lesse River near the Trou des Nutons cave. The mammal assemblage contains a mixture of remains dating from the Late Glacial and Postglacial. The calibrated ages of cut-marked horse bones range from 14 780 cal BP to 15 730 cal BP (S3 Table). Arctic fox remains form 7.1% of the NISP of the Pleistocene assemblage in this cave, whereas red fox remains form more, at 10.7% ([44]; Germonpré unpublished data).

The Trou Magritte cave is about 3 km upstream of the confluence of the Lesse and Meuse Rivers. The Palaeolithic assemblages have been assigned to the Aurignacian and Middle Palaeolithic [51]. The mammal assemblage is primarily composed of remains of horses, reindeer and woolly rhinoceros (*Coelodonta antiquitatis*), some of which bear anthropogenic traces while others show evidence of having been gnawed by cave hyenas [52, 53]. The remains of the foxes constitute 3.1% of the NISP of the Pleistocene assemblage [54]. An AMS date of a mammoth bone indicates that the cave was in use between 46 640 cal BP and 42 970 cal BP (S3 Table).

To follow evolutionary changes in the morphological structure of the lower carnassials in the arctic and red foxes, recent materials were incorporated into the analyses. The contemporary materials included eight sampled populations (545 recent $M_1$) of the red fox from the Palearctic region and seven sampled populations (259 recent $M_1$) of the arctic fox from across its whole arctic range (Fig 1). A list of samples, abbreviations of population names, systematic statuses, collection data and collection names can be found in Table 1. The modern fox samples had been used before in our earlier studies [13, 16, 53, 55]. The number of red foxes in each sample was higher than 50 specimens, whereas the number of arctic foxes in samples ranged from n = 11 (Greenland) to n = 58 (Taymyr). Shape analysis of $M_1$ for each specimen was carried out according to morphotypes defined and illustrated previously for the red fox ([13]; Fig 2). Five groups of morphotypes (K, L, P, R, and S) were used to depict $M_1$ shape variation in the foxes. Morphotypes within each group were identified by Arabian numbers, with higher numbers indicating more complex morphologies. For group P, Szuma [12] classified 5 morphotypes based on numerous variants of crown shape. Definitions of the dental morphotypes used to analyze dental polymorphism in the arctic fox are listed in S4 Table.

For the analysis of fossil materials, both left and right carnassials were used. Previous analyses of frequency distributions of morphotypes in female and male Polish red foxes showed no significant differences between sexes [11]; thus, the sexes in recent samples of the arctic and red foxes were combined. This is very convenient for comparing the ancient and recent fox samples, and does not add error to the statistical analysis.

Frequency distributions of the five morphotype groups were analyzed in ten red and eight arctic fox populations. The significance of interpopulation variation in relation to the dental morphotypes in the foxes was determined with the chi-square tests $\chi^2$. The tooth dimensions: length of $M_1$ ($LM_1$), width of $M_1$ ($WM_1$) and carnivorous index ($LM_1/WM_1$) in relation to the particular morphotypes were verified using MANOVA. One of the authors (ES) has analyzed the sizes and shapes of the lower carnassials of foxes in various other studies [11–13, 37, 55–57].

Before we carried out the multidimensional analyses, we converted the percentage share of each morphotype using angular transformation in conformity with the equation $y = arc\ sin \sqrt{x}$. Angular transformation is needed if the number n on which the individual percentages are based is very variable [58]. The transformations were carried out in MS Excel 97.

Based on the transformed morphotype frequencies of shape variation of $M_1$, we carried out multidimensional scaling (MDS). An agglomeration method was used to determine levels of

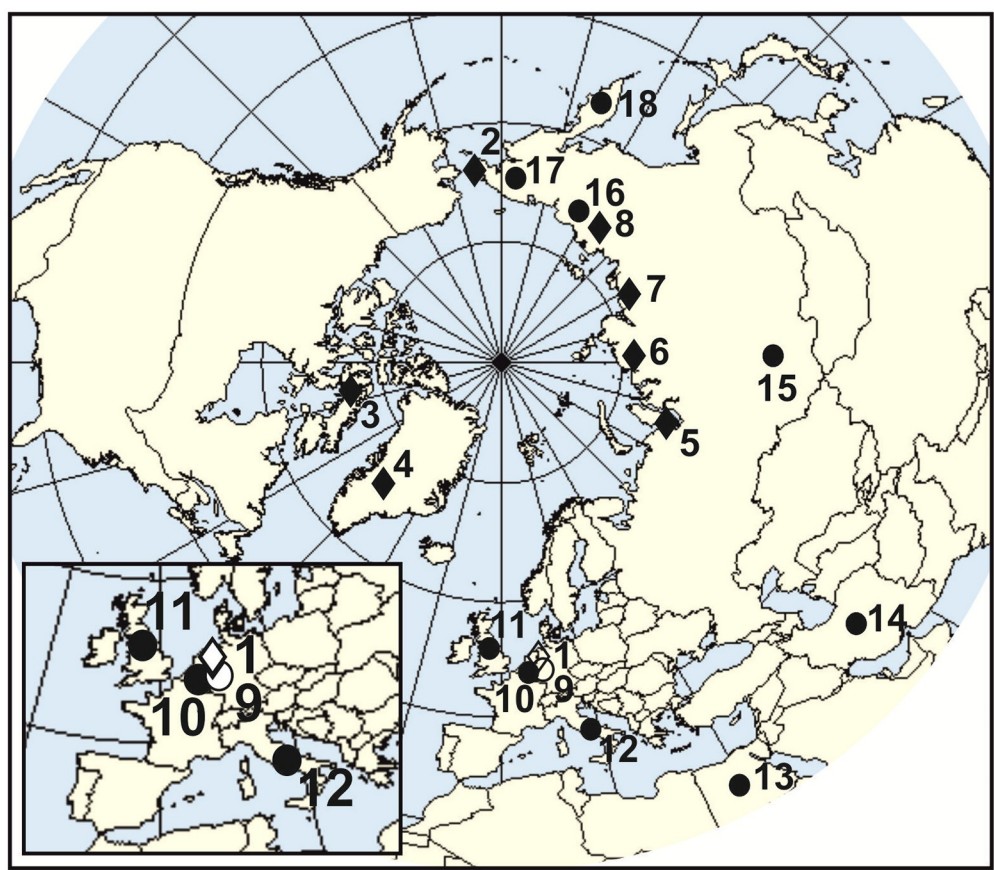

**Fig 1. Geographic distribution of the recent and fossil samples of the arctic *Vulpes lagopus* and red foxes *Vulpes vulpes* used in the study.** Black rhomb–recent populations of the arctic fox measured by Szuma (2011), empty rhomb–fossil populations of the arctic fox measured by Szuma for the study, grey circle–recent populations of the red fox measured by Szuma (2011); empty circle–fossil population of the red fox measured by Szuma for the study. Information about the samples can be found in Table 1.

morphological similarity among recent and fossil populations of the arctic and red foxes. In the agglomeration analysis the city distance (Manhattan) and Ward's method of linking were used. All statistical analyses were performed using Statistica.PL version 9.0.

## Results

### Morphotype frequency in the arctic fox

The distribution of group K morphotypes in fossil arctic foxes from Belgium (pop. 1) was very close to the recent fox population from the Yamal Peninsula (pop. 5, Fig 3)–they had a low share of the morphotype K1 (17% in pop. 1, and 24% in pop. 5), and a presence of the K3 morphotype. The lowest share of morphotype K1 (6%) was observed in arctic foxes from Yakutskaya oblast (pop. 8), whereas the highest was in Beringia (71%, pop. 2), and Greenland (100%, pop. 4). In group L, for the arctic fox we found morphotypes L1 and L2. The frequency of the morphotype L2 in the ancient arctic foxes (pop. 1) was 27%, whereas among the recent populations of the arctic fox the L2 share ranged from 14% in the Yamal Peninsula (pop. 5) to 0% in Beringia (pop. 2). In the ancient sample (pop. 1) we identified four group P morphotypes (P1–P4), and the cumulative share of morphotypes P2, P3, and P4 (55%) was larger than in recent populations of the species. Among the recent foxes only the Baffin Land population (pop. 3)

**Table 1. List of samples of the red and arctic foxes used in the study with specimen numbers, collection times, and sources of the specimens.**

| No | population | n | date/age | collection name |
|---|---|---|---|---|
| | | | the arctic fox | |
| 1 | fossils, Belgium | 45 | 46 640–14 130 BP | RBINS |
| 2 | Beringia | 58 | 1882–1940 | IZ RAS |
| 3 | Baffin Land | 22 | 1951–1999 | NHM |
| 4 | Greenland | 15 | 1973–1988 | ZMA |
| 5 | Yamal Peninsula, Russia | 41 | 1935–1989 | ZMKU; NMNH-P, Kiev; NMNH-Z, Kiev |
| 6 | Taymyr Peninsula, Russia | 60 | 1932–1989 | IZ RAS; ZMSOANSSSR; NMNH-P, Kiev |
| 7 | Kozhevnikov, Russia | 28 | 1938 | IZ RAS |
| 8 | Yakutskaya oblast, Russia | 35 | 1884–1927 | IZ RAS |
| | | | the red fox | |
| 9 | fossils, Belgium | 35 | 46 640–14 130 BP | RBINS |
| 10 | Belgium | 61 | recent | RBINS |
| 11 | England | 70 | 1900–1946 | NHM, London |
| 12 | Italy | 34 | 1878–1976 | NHM, London; Museo di Calci, Piza |
| 13 | Egypt | 89 | 1948–1976 | FMNH, Chicago; NHM |
| 14 | Iran | 80 | 1950–1970 | ZMA, FMNH, USNM |
| 15 | Novosibirskaya oblast, Russia | 133 | 1947–1984 | ZMSOANSSSR |
| 16 | Yakutskaya oblast, Russia | 36 | 1893–1957 | IZ RAS |
| 17 | Chukotka, Russia | 73 | 1892–1973 | IZ RAN, ZM UM |
| 18 | Kamchatka, Russia | 30 | 1841–1934 | IZ RAS; ZMSOANSSSR; NHM |

had a large share of the more complex morphotypes (P3, P4)– 53%. In all other populations of the species the more complicated morphotypes of the transverse cristid of $M_1$ (see Fig 1) had a share lower than 28% (Fig 3).

The distal part of the talonid of the lower carnassials in the arctic fox showed a simplistic occlusal structure. In three populations a small hypocingulid (morphotype R2) appeared at the levels of 2% (pop. 2), 4% – Kozhevnikovo (pop. 7), and 3% (pop. 1) (Fig 3). We did not observe morphotype S2 anywhere across the whole range of the species. In recent and fossil populations of the arctic fox morphotype S1 occurred with a 100% frequency.

## Morphotype frequency in the red fox

The percentage shares of the lower carnassial morphotypes relating to shape of $M_1$ differ between the red and arctic foxes. In the red fox, morphotype K2 oscillated from 64% in Novosibirskaya oblast (pop. 15) to 96% in Iran (pop. 14; Fig 4). In contrast, the share of K1 ranged from 3% (pop. 14) to 34% (pop. 15). In the ancient red foxes (pop. 9), morphotype K1 reached 13%, K2 84%, and K3 just 3%. Group L was represented by all three morphotypes; however, L1 varied from 94% in Belgium (pop. 10) to 57% in Chukotka (pop. 17). In the ancient sample (pop. 9) morphotype L1 constituted 85%, L2 – 11%, and L3 – 4% (Fig 4). Shares of group P morphotypes in the studied samples were characterized by large levels of differentiation. Morphotype P1 oscillated from 0% in Italy (pop. 12) to 30% in pop. 15. The frequency of morphotype P3 in the red fox ranged from 27% (pop. 15) to 47% in Yakutskaya oblast (pop. 16). In red fox populations, the most frequent morphotype was P4 and its values oscillated from 37% (pop. 14) to 61% (pop. 12). The lower carnassials in the red fox were characterized by the presence of two morphotypes, R1 and R2. In population 14, individuals with morphotype R2 comprised 14% of the whole population, whereas in Kamchatka (pop. 18) and Chukotka (pop. 17) red foxes had morphotype R2 frequencies of 81% and 89%, respectively (Fig 4). Among the

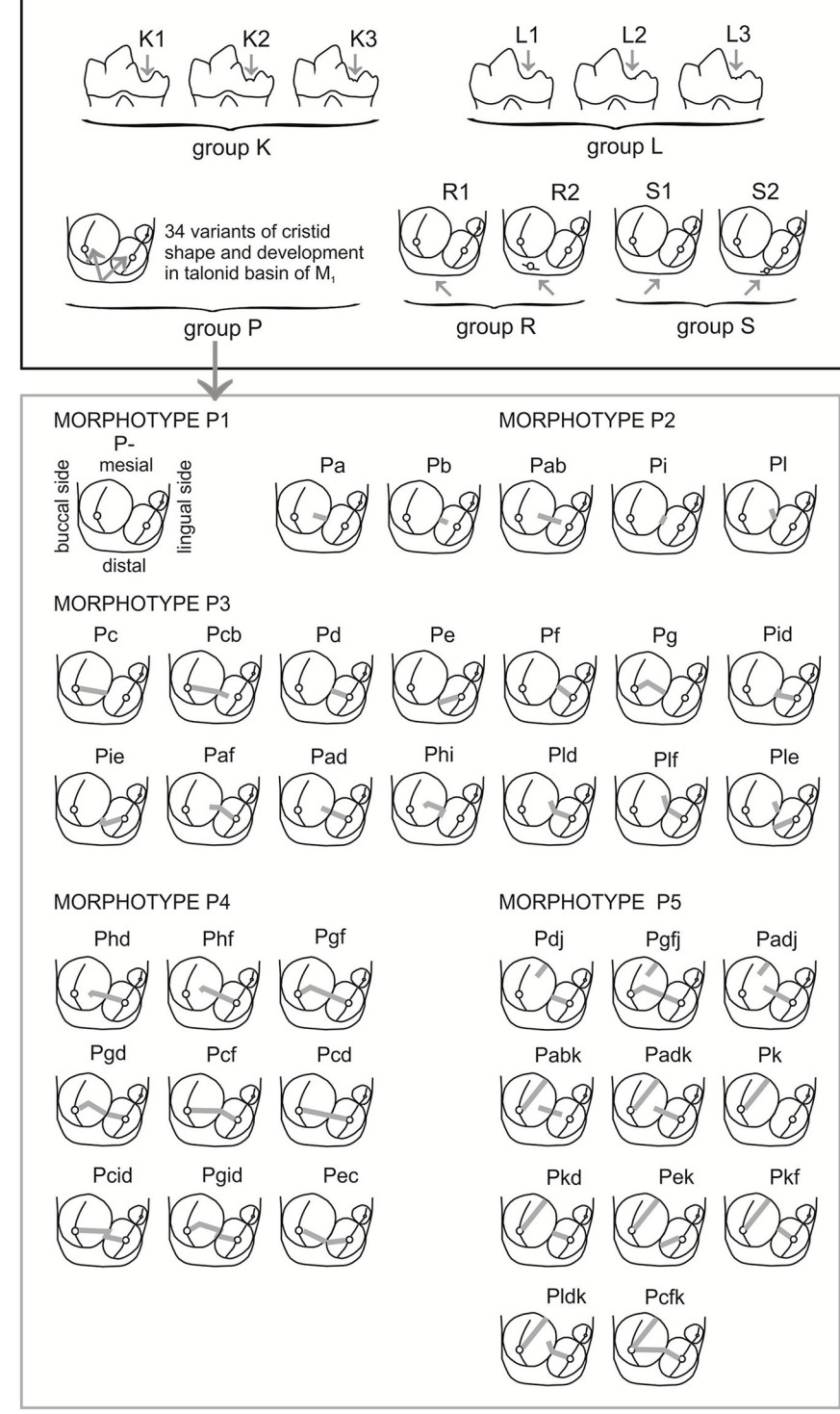

**Fig 2. Graphic depiction of morphological variation of the first lower molar (M₁) in the red and arctic foxes.**
Group K–M$_1$ in lateral view, lingual side; group L–M$_1$ in lateral view, buccal side; group P, R, S–distal part of M$_1$ (talonid) in occlusal view. Definitions of the morphotypes, and morphotype variants are listed in S4 Table.

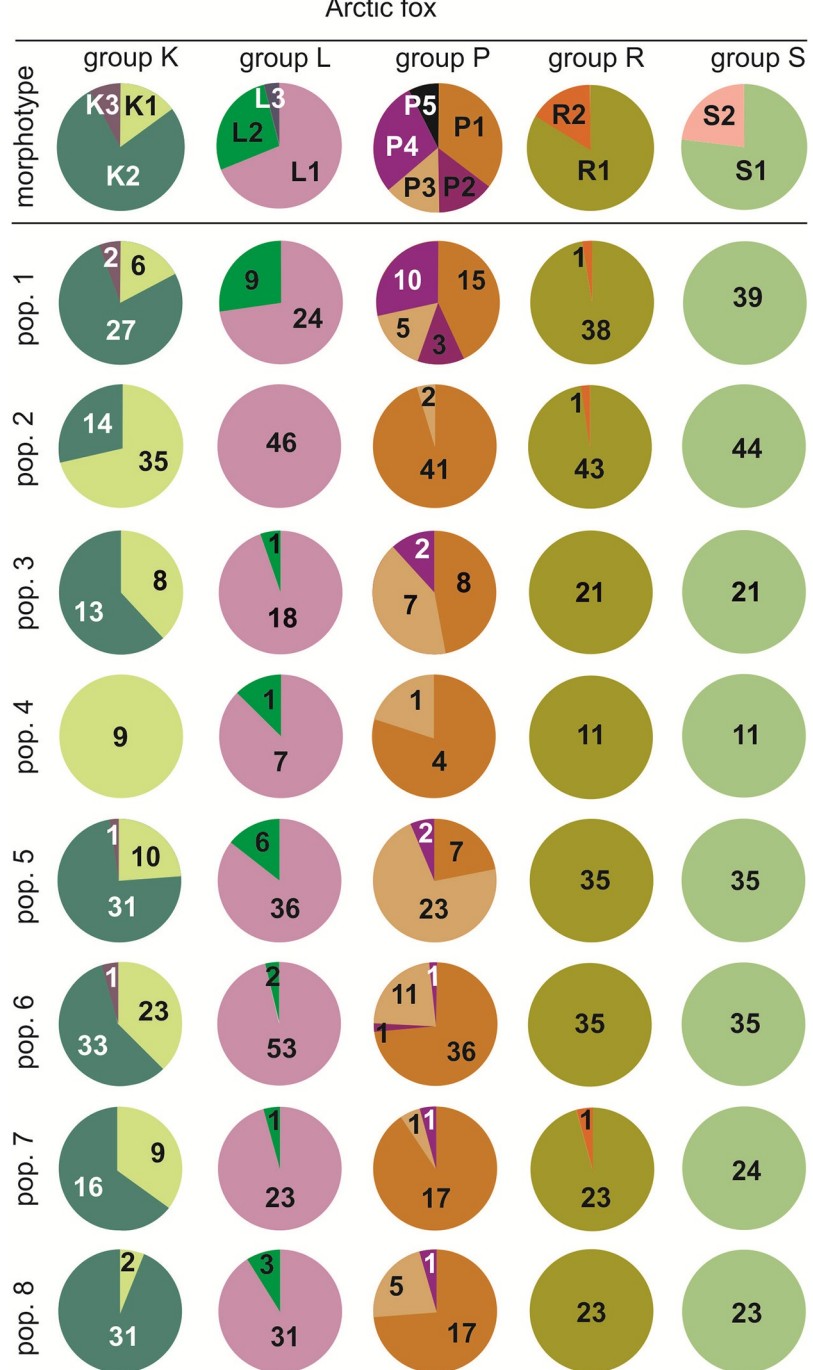

**Fig 3. Share of morphotypes in fossil and recent populations of the arctic fox.** Numerals in each part of the pie chart are the numbers of specimens with the morphotypes. Illustrations of the tooth variants and morphotypes are shown in Fig 2. Information about the samples can be found in Table 1.

fossil red foxes (pop. 9) 50% of individuals possessed the morphotype R1 and the other 50% − R2. Morphotype S2 occurred with lower frequencies than S1 in all populations of the species. The highest levels of the morphotype S2 were found in red foxes from Chukotka (pop. 17; 28%), Kamchatka (pop. 18; 23%), ancient Belgium (pop. 9; 23%), and Italy (pop. 12; 21%),

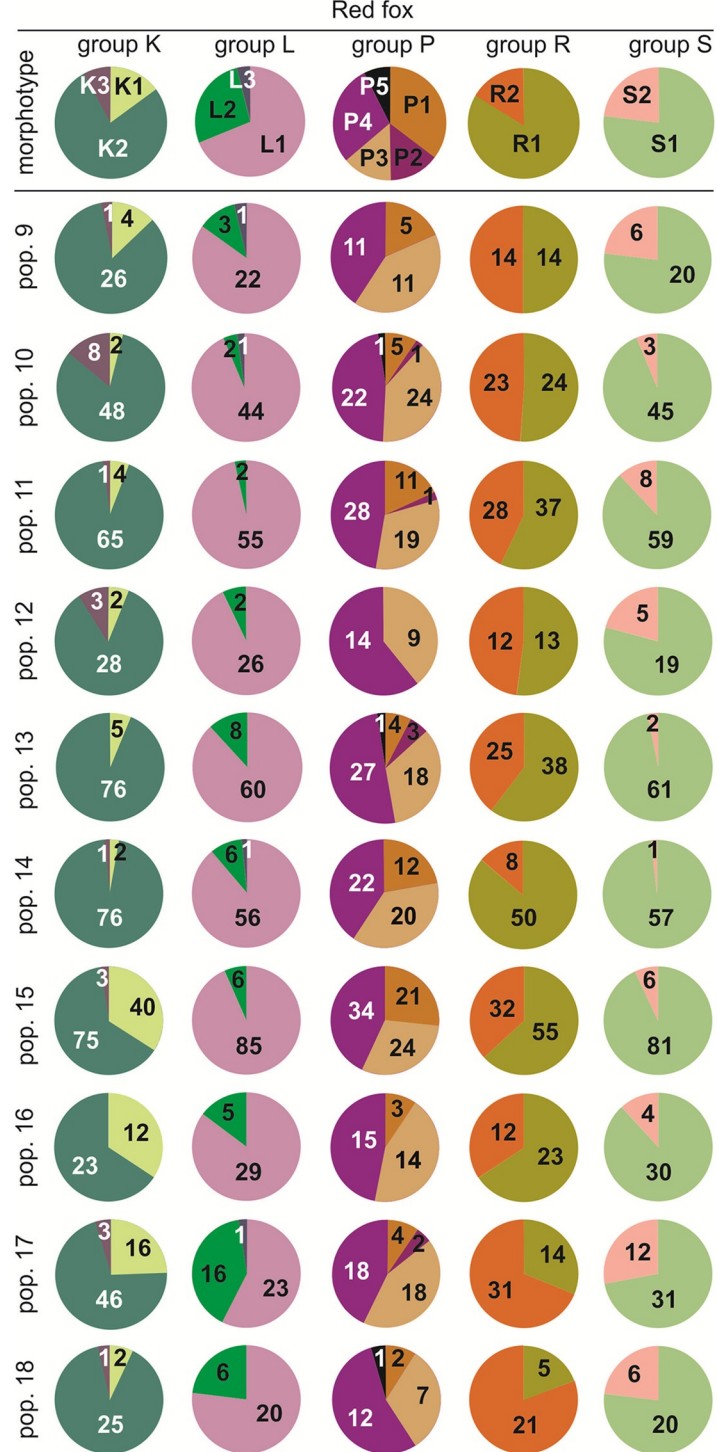

**Fig 4. Share of morphotypes in fossil and recent populations of the red fox.** Numerals in each part of the pie chart are the numbers of specimens with the morphotypes. Illustrations of the tooth variants and morphotypes are shown in Fig 2. Information about the samples can be found in Table 1.

whereas the lowest levels of S2 were in Iran (pop. 14; 2%), Egypt (pop. 13; 3%), and Belgium (pop. 10; 6%).

## Morphotypes versus carnassial size

**Group K.** **The arctic fox**: The median length of the lower carnassials was highest for the K3 morphotype group, and was lower for K2 and K1; however, the differences were not statistically significant (p>0.05; Fig 5). The width of the carnassial crown was also highest in samples with morphotype K3 and lower in the other two (p = 0.005; Fig 5). The carnivorous index

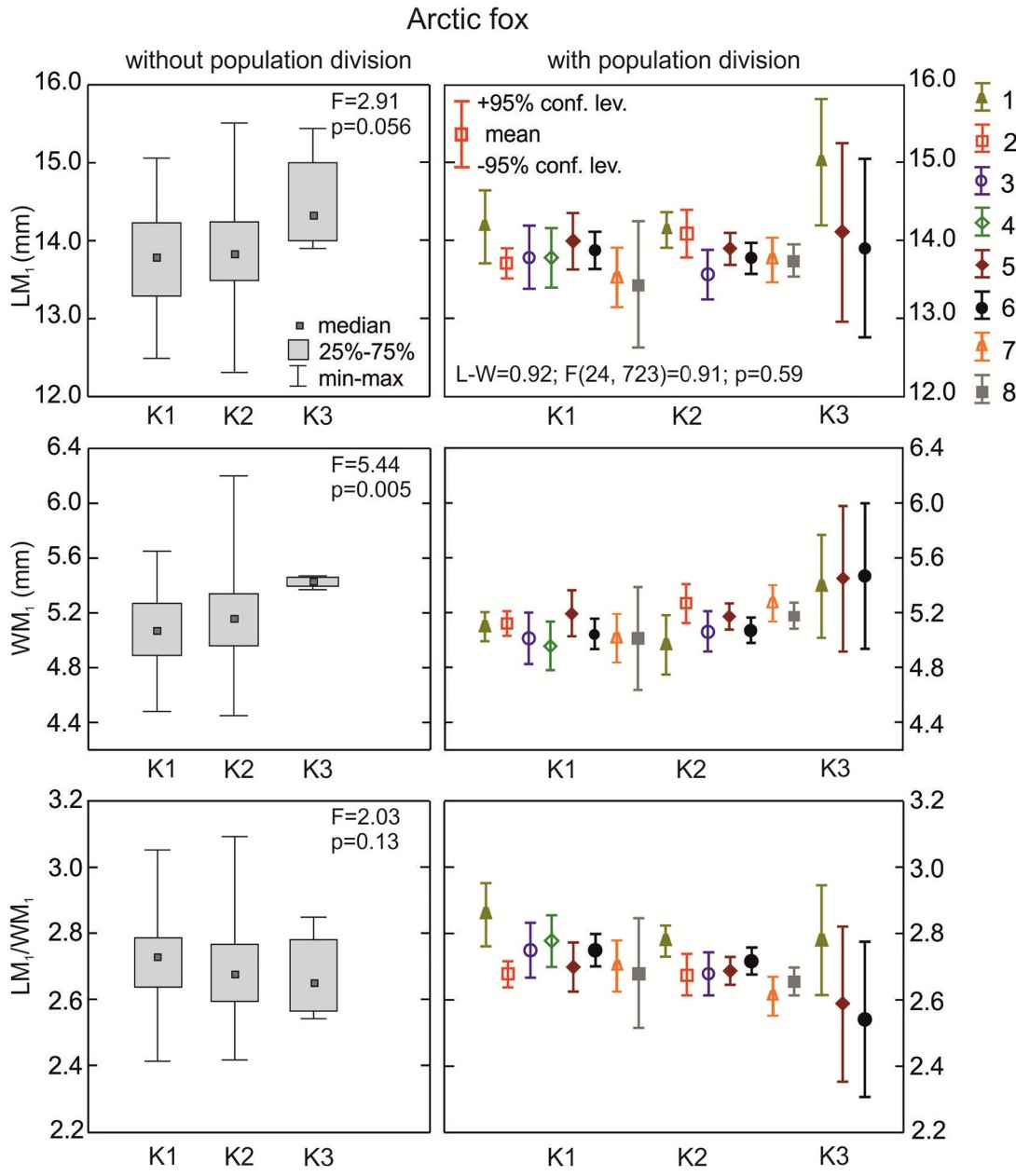

**Fig 5. Comparison of median values for length, width, and size proportion of $M_1$ between various K group morphotypes in the arctic fox with and without population division.** Illustrations of group K morphotypes are shown in Fig 2. Information about the samples can be found in Table 1.

**Table 2. Size variation of the first lower molar ($M_1$) in the arctic and red foxes according to population division (POP), morphotype differentiation (morphotype groups: K, L, P, R, S), and interactions between both factors (POP\*K; POP\*L; POP\*P; POP\*R; POP\*S).** Bold letters indicate statistical significances (p<0.05) of the relationships.

| | | the arctic fox | | | | | | | | | | the red fox | | | | | | | | | |
|---|---|---|---|---|---|---|---|---|---|---|---|---|---|---|---|---|---|---|---|---|---|
| | | $LM_1$ | | | $WM_1$ | | | $LM_1/WM_1$ | | | | $LM_1$ | | | $WM_1$ | | | $LM_1/WM_1$ | | | |
| | | df | F | p | df | F | p | df | F | p | df | F | p | df | F | p | df | F | p | | |
| K | POP | 2 | 2.97 | 0.053 | 2 | 0.50 | 0.609 | 2 | **5.05** | **0.007** | 7 | **9.84** | **0.000** | 7 | **6.91** | **0.000** | 7 | **3.02** | **0.004** | | |
| | K | - | - | - | - | - | - | - | - | - | 1 | 4.31 | 0.038 | 1 | 1.13 | 0.288 | 1 | 1.00 | 0.319 | | |
| | POP\*K | 8 | 1.03 | 0.411 | 8 | 0.64 | 0.745 | 8 | 0.61 | 0.773 | 16 | 0.82 | 0.171 | 16 | 1.02 | 0.431 | 16 | 1.12 | 0.332 | | |
| L | POP | 6 | 2.05 | 0.060 | 6 | **2.46** | **0.025** | 6 | **4.49** | **0.000** | 2 | 1.42 | 0.244 | 2 | **4.97** | **0.007** | 2 | **4.07** | **0.018** | | |
| | L | - | - | - | - | - | - | - | - | - | 1 | 0.34 | 0.558 | 1 | 1.12 | 0.290 | 1 | **5.06** | **0.025** | | |
| | POP\*L | 6 | 0.71 | 0.641 | 6 | **2.22** | **0.041** | 6 | **2.62** | **0.017** | 11 | 1.09 | 0.368 | 11 | 1.29 | 0.225 | 11 | 0.93 | 0.515 | | |
| P | POP | 1 | 2.39 | 0.124 | 1 | 3.38 | 0.067 | 1 | 1.04 | 0.309 | 3 | **24.03** | **0.000** | 3 | **16.82** | **0.000** | 3 | **4.84** | **0.002** | | |
| | P | 1 | 0.00 | 0.995 | 1 | 1.64 | 0.201 | 1 | 3.30 | 0.071 | 1 | 0.07 | 0.788 | 1 | 0.04 | 0.832 | 1 | 0.03 | 0.866 | | |
| | POP\*P | 13 | 1.03 | 0.426 | 13 | 1.33 | 0.196 | 13 | **1.79** | **0.047** | 20 | **1.80** | **0.018** | 20 | **2.26** | **0.002** | 20 | **1.71** | **0.030** | | |
| R | POP | 2 | 1.35 | 0.261 | 2 | 1.06 | 0.348 | 2 | 0.44 | 0.643 | 9 | **40.8** | **0.000** | 9 | **18.98** | **0.000** | 9 | **7.60** | **0.000** | | |
| | R | - | - | - | - | - | - | - | - | - | 1 | **9.5** | **0.002** | 1 | **6.85** | **0.009** | 1 | 0.00 | 1.000 | | |
| | POP\*R | 2 | 0.52 | 0.597 | 2 | 1.72 | 0.180 | 2 | 0.88 | 0.415 | 9 | 1.4 | 0.198 | 9 | **2.53** | **0.008** | 9 | 1.90 | 0.052 | | |
| S | POP | - | - | - | - | - | - | - | - | - | 9 | **12.91** | **0.000** | 9 | **5.72** | **0.000** | 9 | **4.74** | **0.000** | | |
| | S | - | - | - | - | - | - | - | - | - | 1 | **9.03** | **0.003** | 1 | **7.25** | **0.007** | 1 | 0.05 | 0.824 | | |
| | POP\*S | - | - | - | - | - | - | - | - | - | 9 | 1.03 | 0.414 | 9 | **2.13** | **0.026** | 9 | 1.54 | 0.131 | | |

of arctic fox samples with morphotypes K1, K2 or K3 did not vary significantly (p>0.05; Fig 5). The MANOVA for the arctic fox was only carried out for a few samples because of low n in the samples and morphotype fractions. However, there were statistically significant differences in $LM_1/WM_1$ between three populations with various group K morphotypes (p = 0.007; Table 2).

**The red fox**: The lower carnassials in red foxes with morphotype K1 had the highest median crown length (Fig 6). There was a statistically significant difference in the median values for $LM_1$ between foxes with morphotypes K1, K2, and K3 (p = 0.001). Morphotype K3 was characterized by a moderate median carnassial crown length. Similarly, the median width of the lower carnassials was highest in group K1, moderate in K3, and lowest in K2 (Fig 6). With regards to the carnivorous index ($LM_1/WM_1$), red foxes with the various K morphotypes did not show any significant differences (p>0.05) in the central distribution measure (Fig 6). However, when taking into account the population division of the samples with the various K morphotypes, there were statistically significant differences in all the $M_1$ parameters (Table 2).

**Group L.** **The arctic fox**: Samples with morphotypes L1 and L2 had similar median values of $WM_1$ and carnivorous index ($LM_1/WM_1$) (p>0.05; Fig 7). Medians for crown length of the lower carnassials were higher in morphotype L2 than in L1 (p<0.05). There were statistically significant differences (p<0.001) between some populations with the various L morphotypes, as calculated with the $LM_1/WM_1$ index. When considering the population division of samples with L morphotypes, there was statistically significant variation in $WM_1$ and $LM_1/WM_1$ between groups of arctic foxes (Table 2).

**The red fox**: Medians for $LM_1$, $WM_1$, and $LM_1/WM_1$ showed similar values in L1, L2, and L3 morphotype (p>0.05; Fig 8). However, when considering population and L group divisions there were statistically significant differences in the means of $WM_1$ and $LM_1/WM_1$ between samples (p<0.05; Table 2).

**Group P.** **The arctic fox**: Group P morphotypes were similar (p>0.05) in all metrical parameters and had similar carnivorous indices (Fig 9). When considering population and

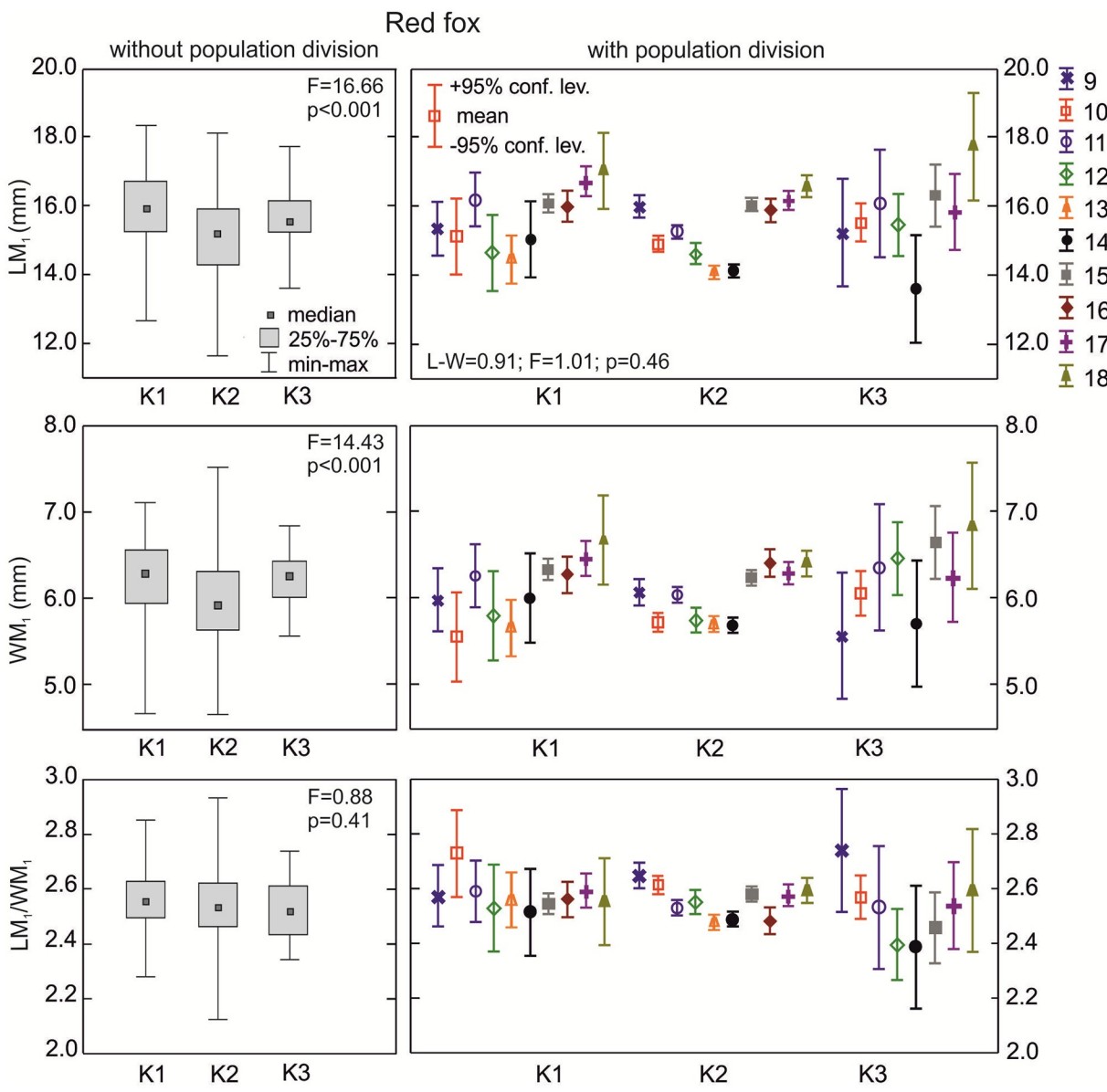

**Fig 6. Comparison of median values for length, width, and size proportion of M$_1$ between various group K morphotypes in the red fox with and without population division.** Illustrations of group K morphotypes are shown in Fig 2. Information about the samples can be found in Table 1.

morphotype division (P1–P4), a significant cumulative effect of both factors on LM$_1$/WM$_1$ between sample variation (p<0.05; Table 2) was found.

**The red fox**: Group P morphotypes had similar (p>0.05) medians of metrical parameters and carnivorous indices (Fig 10). However, when considering differentiation between populations, significant variation (p<0.005) was observed. The MANOVA procedure confirmed there was statistically significant variation for all parameters of M$_1$ between populations with various P morphotypes (Table 2). The cumulative effect of population and morphotype division translated into significant variation in the size and size proportion of M$_1$ between samples.

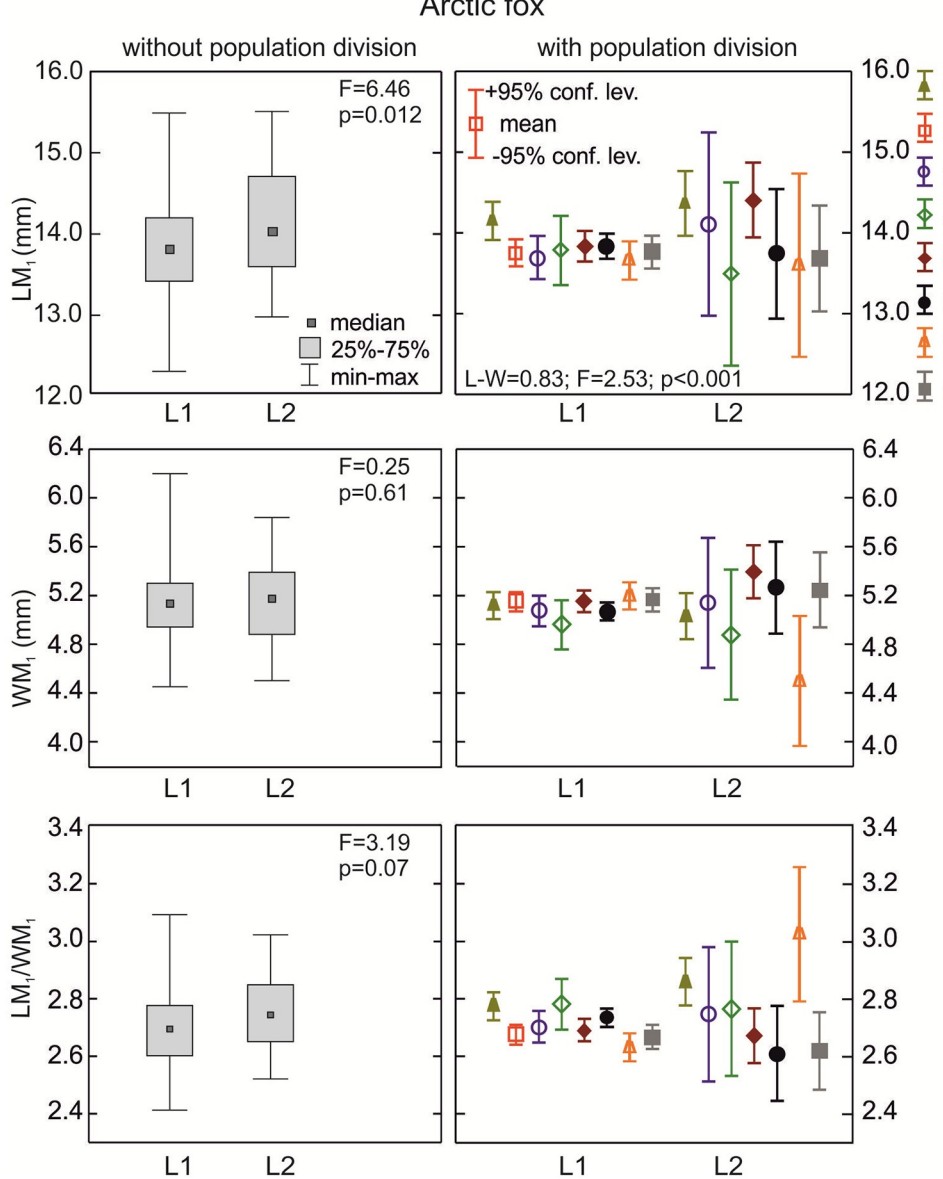

**Fig 7. Comparison of median values for length, width, and size proportion of $M_1$ between various group L morphotypes in the arctic fox with and without population division.** Illustrations of group L morphotypes are shown in Fig 2. Information about the samples can be found in Table 1.

**Group R.** **The arctic fox**: Median length and width were significantly larger in the arctic foxes with the R2 morphotype than those with R1 (Fig 11). Samples with morphotypes R1 and R2 had similar (p>0.05) carnivorous indices (Table 2). The cumulative effect of both factors is difficult to verify because the *df* is only 2 (Table 2).

**The red fox**: Specimens with morphotypes R1 and R2 had significant differences (p<0.001) in the median values of length and width of $M_1$ (Fig 11). Only the median of carnivorous index was similar between the morphotypes (p>0.05; Fig 11). When considering population and R morphotype divisions, all metrical parameters of $M_1$ showed highly significant variation (p<0.001) between samples. The cumulative effect of the population and R division was visible in $WM_1$ (Table 2).

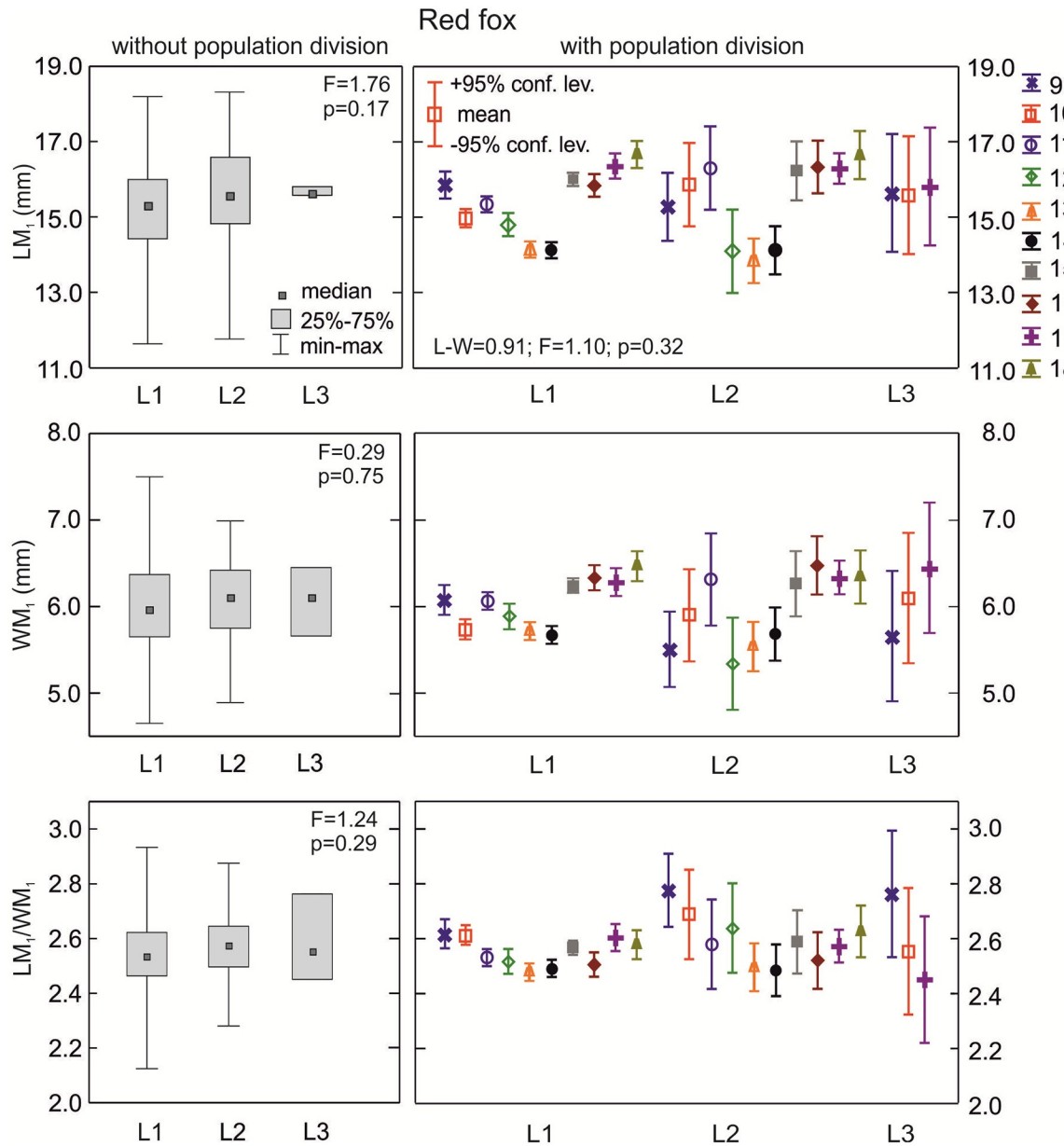

**Fig 8. Comparison of median values for length, width, and size proportion of M₁ between various group L morphotypes in the red fox with and without population division.** Illustrations of group L morphotypes are shown in Fig 2 Information about the samples can be found in Table 1.

**Group S.** **The arctic fox**: The S1 morphotype occurred with 100% frequency.

**The red fox**: Specimens with morphotype S2 had higher median values of metrical characters than those with morphotype S1 ($p<0.001$; Fig 12); the $LM_1/WM_1$ index was also higher in those with morphotype S2 than those with morphotype S1, based on the level $p = 0.05$ (Fig 12). The MANOVA procedure found statistically significant ($p<0.001$) differentiation in $LM_1$, $WM_1$, and $LM_1/WM_1$ in population and morphotype separated red foxes. The cumulative effect of the population and S division was visible in $WM_1$ ($p<0.05$; Table 2).

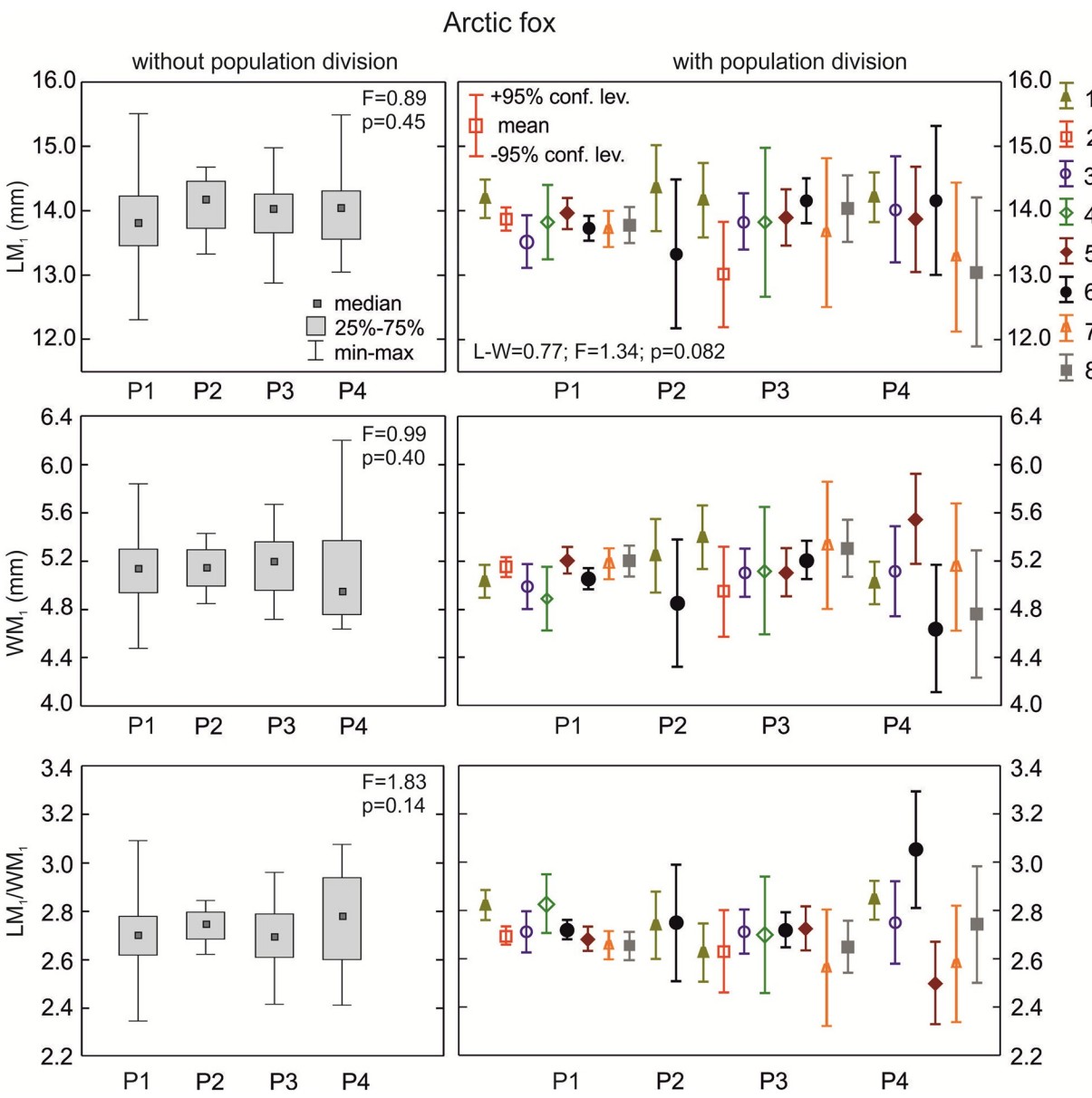

**Fig 9. Comparison of median values for length, width, and size proportion of M₁ between various group P morphotypes in the arctic fox with and without population division.** Illustrations of group P morphotypes are shown in Fig 2. Information about the samples can be found in Table 1.

## Morphological distances between populations

In the arctic fox, the MDS result, based on the angularly transformed frequencies of $M_1$ morphotypes, suggested that the ancient sample (pop. 1) was distant compared with modern samples of the species. The Beringia and Greenland populations of the arctic fox (pops 2, 4; Fig 13) were very different from both fossil and all modern samples.

The red fox had smaller morphological distances between fossil and modern foxes than the arctic fox. The fossil Belgian red foxes (pop. 9) were closest to recent foxes from Belgium (pop. 10), Italy (pop. 12), England (pop. 11), and Egypt (pop. 13), and most distant to populations from Kamchatka and Chukotka (pops 18, 17; Fig 13).

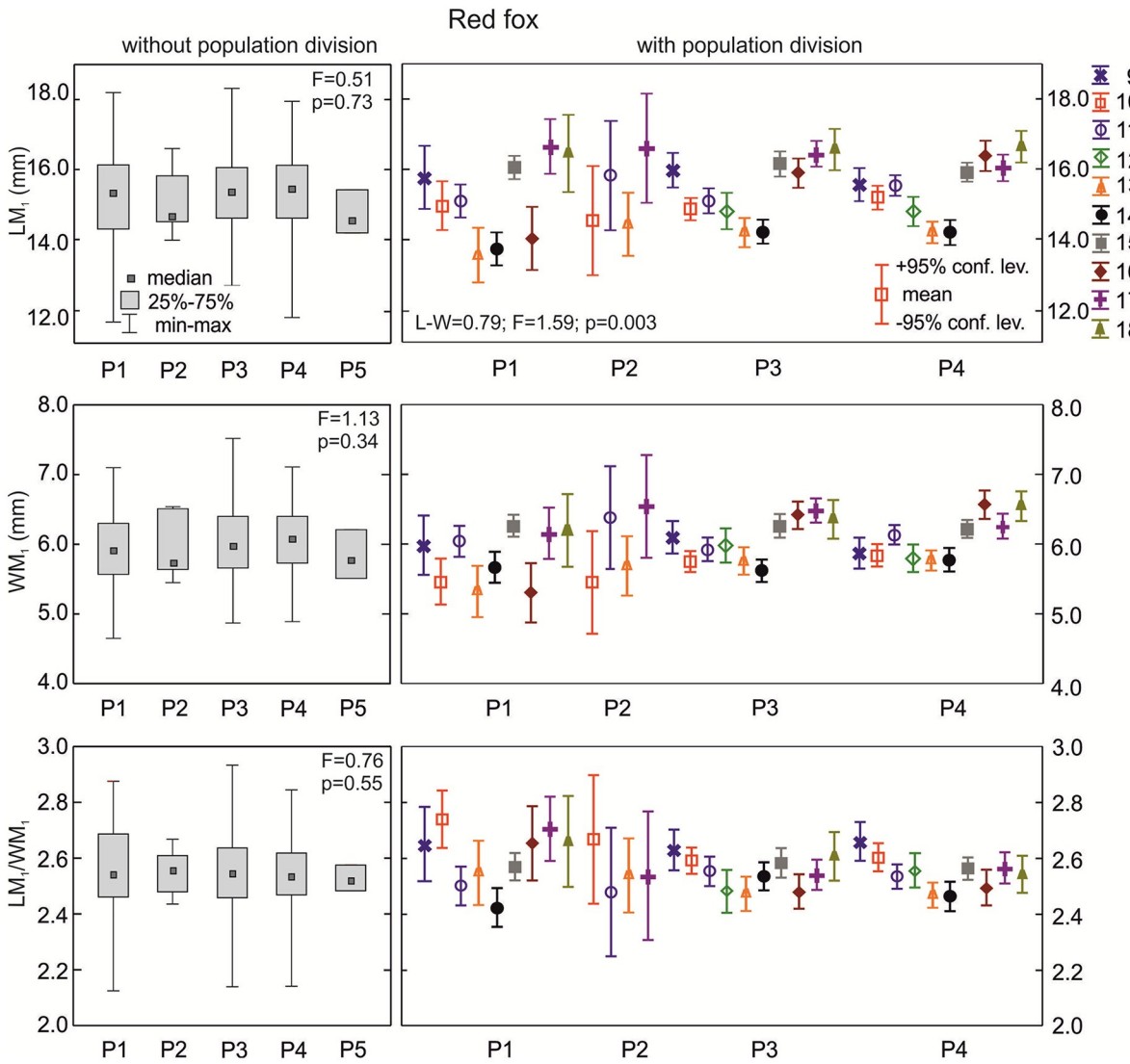

**Fig 10. Comparison of median values for length, width, and size proportion of M₁ between various group P morphotypes in the red fox with and without population division.** Illustrations of P group morphotypes are shown in Fig 2. Information about the samples can be found in Table 1.

The cluster tree based on the angularly transformed frequencies of M₁ morphotypes separated the red and arctic foxes very well (Fig 14).

## Discussion

### Lower carnassial evolution in the arctic fox

The current knowledge on the evolutionary history of *Vulpes* suggests that the genus appeared ca. 7 Ma BP [59]; the oldest fox remains have been found in the late Miocene deposits of Africa, Chad–*Vulpes riffautae* [59]. The arctic fox probably evolved from an ancestral line of a cold adapted ancestor–an early Pliocene (3.60–5.08 Myr ago) fox, *Vulpes qiuzhudingi*, from the Tibetan Plateau [60, 61]. This early arctic fox from the Himalaya (holotype: IVPP V18923) was characterized by very sharp, hypercarnivorous teeth. In this specimen, the talonid of the

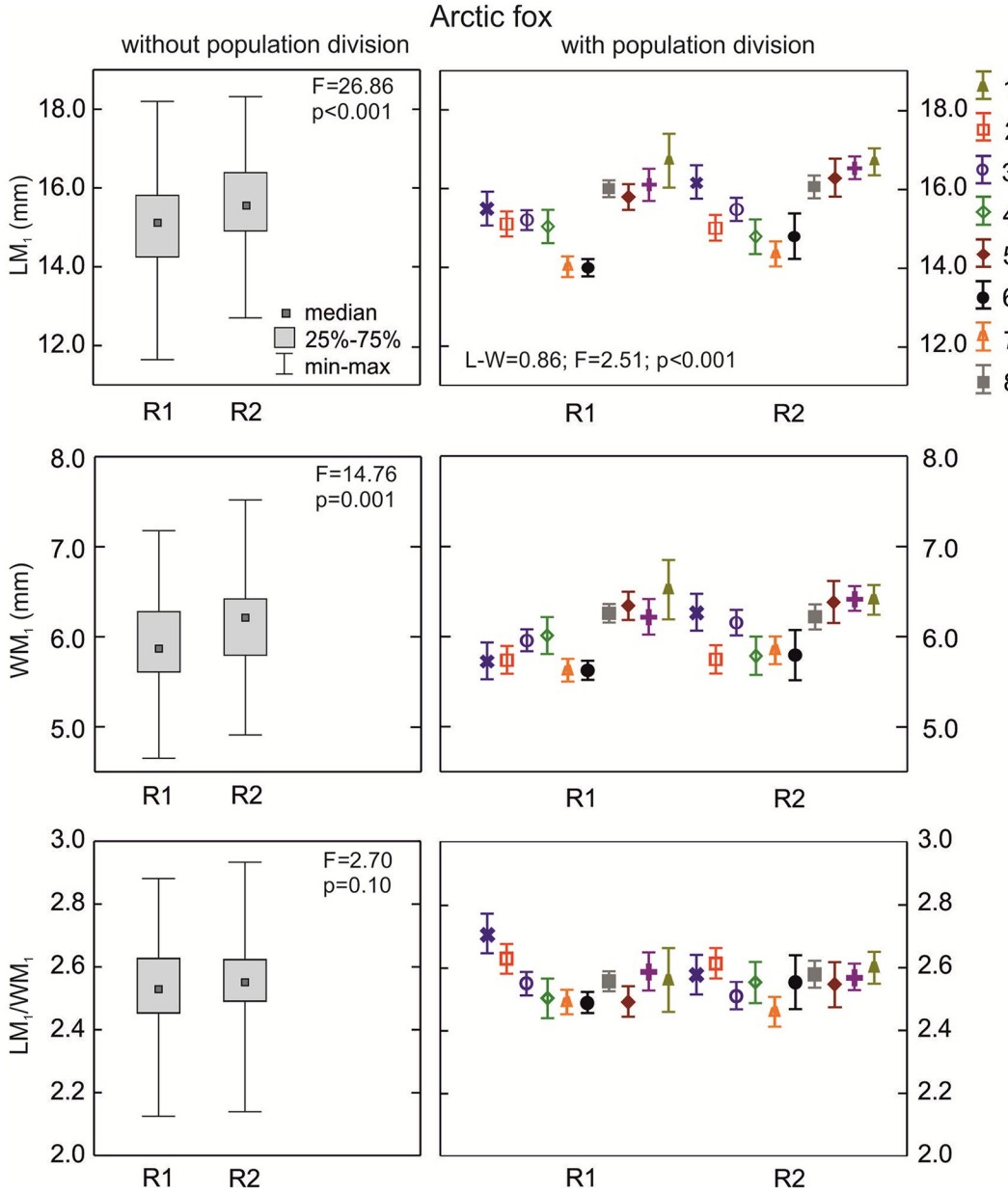

**Fig 11. Comparison of median values for length, width, and size proportion of M$_1$ between various group R morphotypes in the red fox with and without population division.** Illustrations of group R morphotypes are shown in Fig 2. Information about the samples can be found in Table 1.

first lower molar was narrow, with a tendency to be distally narrower, and its second lower premolar had a relatively narrow crown. The shape of the carnassials had a primitive character, exemplified by the presence of additional small conules (a postmetaconulid and entoconulid) in the concavity between the metaconid and entoconid–morphotype K3. The Pleistocene arctic foxes from Belgium had a low share of K1 morphotypes (17%; an absence of small conule/-s in the concavity between the metaconid and entoconid). The more complicated shapes K2, K3, and L2 had a larger share in the fossil arctic foxes than in the recent ones. Moreover, the occlusal surface of the talonid on M$_1$ in the fossil arctic foxes from the Late Pleistocene of

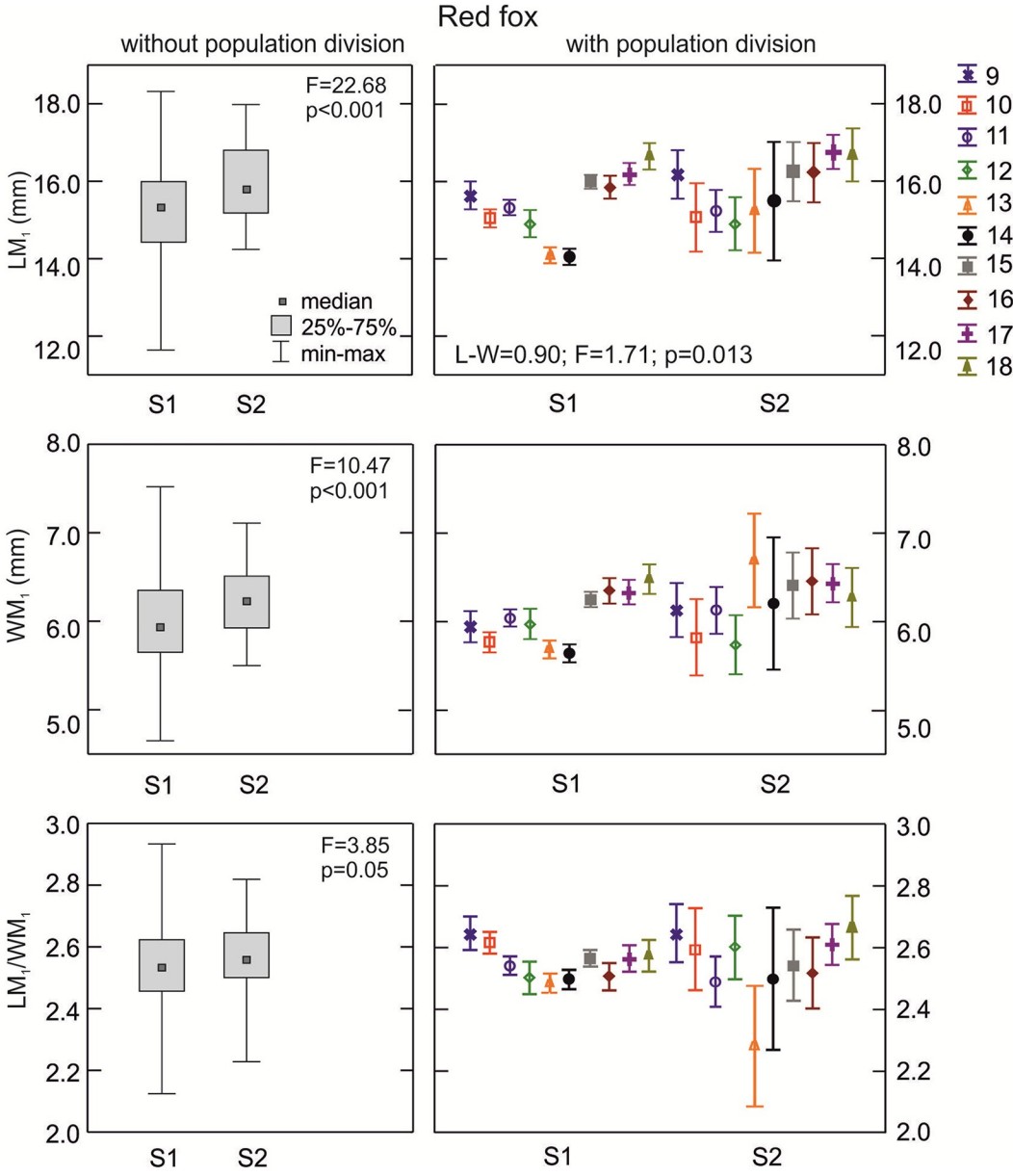

**Fig 12. Comparison of median values for length, width, and size proportion of $M_1$ between various S group morphotypes in the red fox with and without population division.** Illustrations of group S morphotypes are shown in Fig 2. Information about the samples can be found in Table 1.

Belgium had more complicated enamel cristids in the central part of the talonid basin than the recent arctic foxes. The comparison of the morphotype characterization of the lower carnassials between the ancient arctic foxes from Belgium, the ancient holotype from the Tibetan Plateau, and individuals from geographically distant recent populations revealed that the arctic fox has undergone distinctive, parallel evolutionary changes in terms of tooth shape, simplification of crown shape and disappearance of some primitive characters, while also undergoing changes in the size and size proportions of the carnassials [62]).

The newly described, oldest arctic fox *V. qiuzhudingi* was bigger than both the late Pleistocene foxes from Belgium, and the recent arctic foxes from across their whole range. This new

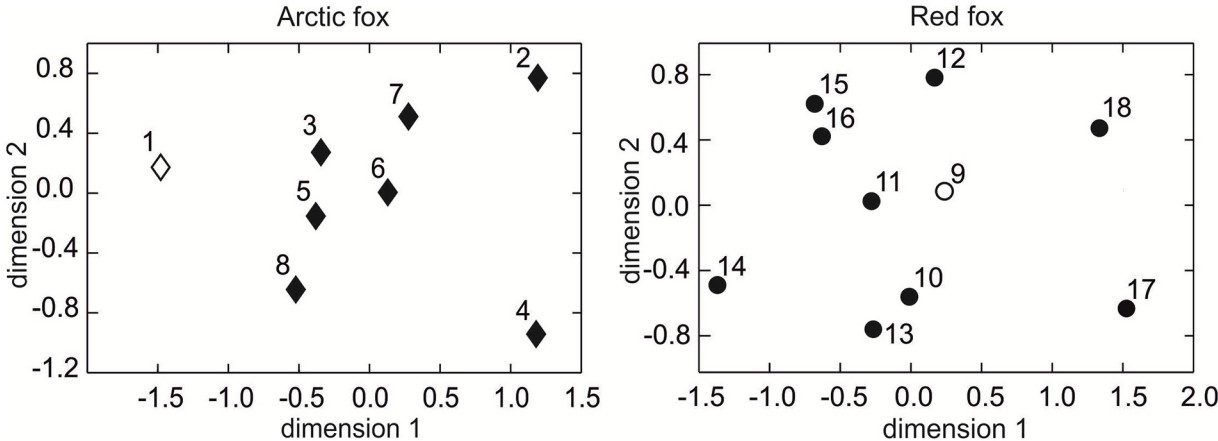

**Fig 13. Distribution of the populations of the arctic and red foxes in a two-dimensional plane based on the morphotype variations of the first lower molar.** Black rhomb–recent populations of the arctic fox, empty rhomb–fossil populations of the arctic fox, grey circle–recent populations of the red fox, empty circle–fossil population of the red fox. Information about the samples can be found in Table 1.

species of arctic fox displayed great predatory adaptations in its dental system because of its much smaller talonids and the presence of several primitive characters in the lower carnassials. The large gaps between the Early Pliocene arctic fox from the Tibetan Plateau and Late Pleistocene arctic fox from Belgium, and then between the latter and recent arctic fox could suggest intensive evolutionary processes have been taking place since the first occurrence of the arctic fox progenitor. The heavy hypercarnivorous specialization of the new species, *V. qiuzhudingi*,

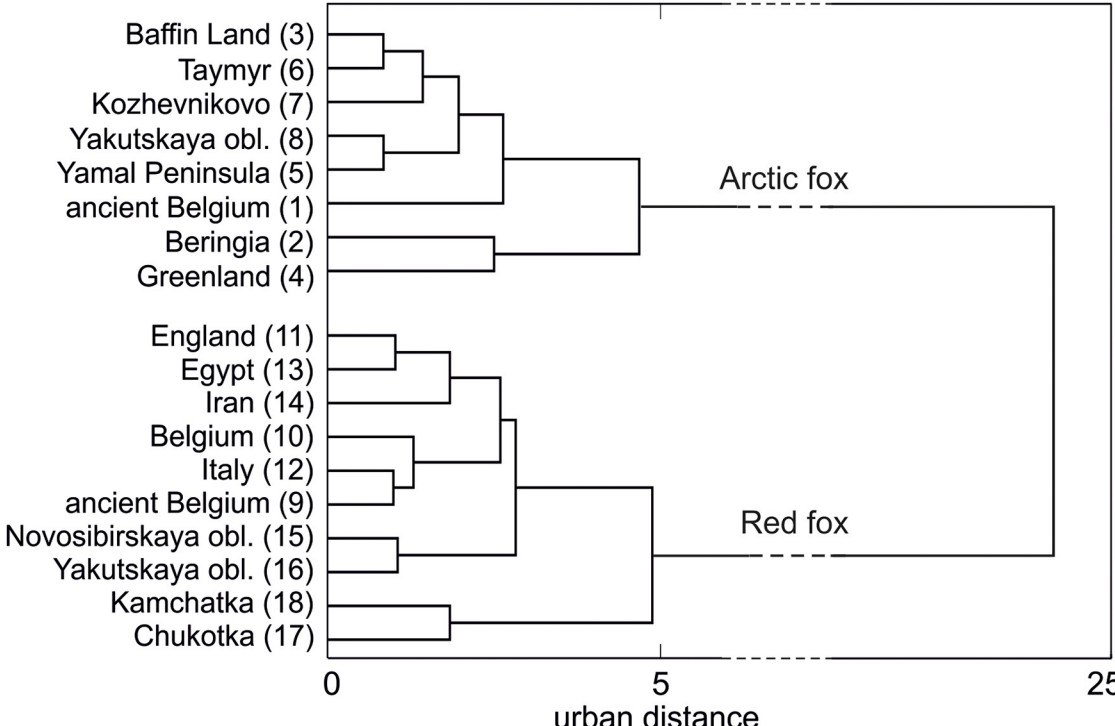

**Fig 14. Agglomeration of the recent and fossil populations of the arctic and red foxes constructed by the Ward method and Manhattan distances using the percentage shares of the morphotypes of the first lower molar.**

was linked with its adaptation to the very cold conditions present in the Himalayan region at that time, and the presence of sympatric cold-adapted mammals (large body size, long hair, snow sweeping structures)–a potential food base for foxes. A new hypothesis was recently formulated suggesting that the paleo-elevation of the Tibetan Plateau was the 'starting line' for most Quaternary Ice Age fauna [61]. The oldest representatives of the woolly rhino *Coelodonta thibetana* have been identified from the fossil beds of the Tibetan Plateau [62]. Deng et al. [61, 63] indicated that *C. thibetana* was the baseline form in the woolly rhino lineage history. Scavenging of the then Tibetan foxes *V. qiuzhudingi* on carcasses of big, cold-adapted forms of mammals required the relevant body and tooth construction properties. The crown length of the lower carnassials of the ancestral form of the arctic fox reached 16.6 mm, whereas among the recent arctic foxes the specimen with the largest length of this molar– 15.51 mm–was recorded in the Yamal Peninsula population, with the whole sample having a mean of 13.74 mm [55]. In the fossil arctic foxes from the Late Pleistocene of Belgium the largest length of $M_1$ was 15.49 mm, whereas the mean of $LM_1$ was 14.21 mm. In the evolutionary history of the arctic fox we can observe a progressive reduction in tooth size and an undermining of carnivorous specialization in the dental system. Considering, the great gap between the progenitor of the arctic fox line–*V. qiuzhudingi*, the next step of the *V. lagopus* from the Late Pleistocene, and then later the recent arctic fox–the magnitude of the changes in the molar's structure and size fit the length of the time intervals between the studied evolutionary stages. We can observe a spectacular change in the molar's morphology from the Pliocene to the Late Pleistocene (about 4 Myr BP) and then from the Late Pleistocene to the recent Holocene (about 42–12 kya BP).

### Lower carnassial evolution in the red fox

The recent red fox is the largest of the extant representatives of the *Vulpes* genus and occupies the largest natural range with a very wide spectrum of habitats. The species has a very malleable genotype that has enabled it adapt to rapid changes in recent times. With great success the red fox has inhabited a new ecological niche–landscapes heavily transformed or occupied by man. The fox's great plasticity means that the species is very expansive, leading to it occupying even the northernmost areas of the globe [64, 65]. The northern expansion of the red fox in Sweden over the last century is a consequence of the population decrease of the grey wolf and European lynx, and increasing ungulate densities. Thus the red fox has expanded into the native range of the arctic fox and competes with it effectively [66]. This evolutionary success of the red fox is most probably a result of its very flexible phenotype, e.g., inter alia, its tooth morphology allows it to use many different types of food; the red fox has 15 morphotypes of crown structure of the lower carnassial, whereas the arctic fox has just 12. The main indicator of the lower carnassials of the red fox is the larger significance of the talonid, i.e. the presence of an entostylid, and a more developed transverse cristid than the arctic fox. Among the worldwide population of the red fox, we have previously observed 34 variants in the form of the transverse cristid in the talonid basin [14], whereas in the arctic fox only 14 [16].

The ancient sample of the red fox from the Late Pleistocene of Belgium had smaller differences in the lower carnassial morphology compared to modern samples of the species, than those observed between the arctic fox samples. Only for group R and S morphotypes we found significant differentiation in the size of the first lower molar ($p < 0.05$). We observed that the red fox specimens with additional cusps present in the most distal part of the lower carnassials (R2, S2 morphotypes) tend to have longer or wider teeth, e.g. the Late Pleistocene of Belgium, Chukotka, and Kamchatka red foxes. It seems that the presence of the additional distal cusps on $M_1$ could have been affected by the threshold mechanism. For some characters to be

expressed during ontogeny (i.e for some structures and their elements to occur), they need to reach or exceed an appropriate threshold value (size, concentration) during the early stages of development [21, 67]. Additional cusps on the anterior or distal parts of the tooth crown are specific to ancestral, primitive carnivorous mammals [68]. Over the course of red fox's evolutionary history, the small cusps on the talonid have occurred at a relatively stable frequency.

The baseline and most primitive set of dental characters in the red fox have been found in the southern regions of Asia (India, Turkmenistan, Morocco, Amurskaya oblast), indicating that the *V. vulpes* line most probably radiated in Asia Minor, Middle Asia, and India [13]. For example, the frequency of occurrence of the entoconulid (K2, K3 morphotypes) is highest in recent red foxes from Morocco, Saudi Arabia, Iran, Turkmenistan, Pakistan, India, and Futsing province in China [13]. In a specimen of *V. riffatuae* from the Late Miocene of Chad, the entoconulid was not found due to heavy wearing of the occlusal surface of the $M_1$ [59]. However, among other ancient foxes, e.g. in *Vulpes beihaiensis*, a well shaped entoconulid was confirmed [69], while in *Vulpes praecorsac* an entoconulid was found on one of the three first lower molars [68]. The presence of entoconulids in some specimens of ancient foxes (*V. beihaiensis*, *V. praecorsac*, *V. vulpes*: pop. 9) and comparable frequencies of additional cusps in the occlusal structures of $M_1$ in ancient and recent red foxes indicated there was no substantial shift in the morphology of the tooth from the Pleistocene to the recent time. Some changes in tooth morphology toward hypocarnivory were observed in the Polish red foxes over seventy years of the 20th century [70]. Microevolutionary changes in the occlusal structure of the premolars (i.e. $P^4$, $P_3$, $P_4$) and the incisor ($I^3$) were found, whereas no relevant trends were detected in the other cheek teeth of the species [70]. The evolutionary history of the Carnivore group has revealed that less specialized forms (e.g. Miacids, recent Canids) retained a greater ability to react to environmental changes than heavily specialized ones (e.g. ancient Hyaenodontidae, Nimravidae). Over the course of evolution, the dental pattern of the red fox has oscillated around a certain norm, so that at any given point in the fox's phylogeny, it could react effectively to new factors and environmental changes. Red fox teeth in modern populations are an example of a prompt evolutionary response of tooth morphology to dietary changes [70]. This morphological shift has confirmed that the *Vulpini* group has great abilities to adapt to fast-changing environments.

## Phylogeny of the arctic and red foxes

The morphological patterns of the lower carnassials showed very clear separation between populations representing the arctic and red fox lineages. The percentage shares of particular crown shape morphotypes evidently separated the fox species, and their geographically and chronologically distant populations. In the phenogram we constructed, the Beringian and Greenland recent arctic foxes made a separate branch. The Siberian arctic foxes (Kozhevnikovo, Taymyr, Yamal, Yakutsk) and the population of Baffin Land were the most similar morphologically, whereas the population of ancient arctic foxes from Belgium was more different to recent samples. Another scenario was observed between red fox populations, where the ancient red foxes from the Late Pleistocene of Belgium were morphologically very close to the recent ones from Italy and Belgium. The most morphologically distant to other populations were the recent red foxes from Kamchatka and Chukotka.

The phylogenetic tree based on the morphological characteristics of $M_1$ for the arctic fox to some extent agreed with a tree based on metric parameters [62]. Both studies indicate fossil arctic foxes to be distant to recent populations of the species. This evolutionary line of the arctic fox occupied Western Europe during cold periods of the Pleistocene, and after the retreat of the last glaciers arctic foxes from Belgium did not go north. Dalén et al. [71] based on

ancient DNA analyses suggested that European fossil populations of the arctic fox most probably died out with the end of the Pleistocene epoch. The morphological analysis of the lower carnassials indicated that the arctic foxes from the Late Pleistocene of Belgium shared the same fate as other foxes populating glacier free-areas in western and southern Europe during the Late Pleistocene.

A comparison of the cladograms for the red fox populations based on morphotype variation and metric characteristics of the lower carnassials showed that they uniformly indicated that the fossil red foxes from the Pleistocene of Belgium are various to recent foxes. In the size of $M_1$ and proportion index ($LM_1/WM_1$) ancient Belgian foxes were similar to recent foxes from Siberia, i.e. Novosibirskaya oblast, Yakutskaya oblast, Chukotka, Kamchatka [62]. This was clear because in carnivores very high correlations between length of the $M_1$ and body mass ($r_{25} = 0.97$, p<0.001), according to Flower [72] has been reported before. The red fox follows Bergmann's rule, so Siberian foxes are much bigger than red foxes from the southern areas of their range [73].

Based on the morphotype patterns of the lower carnassials we revealed that ancient Belgian red foxes were most similar to recent Belgian and Italy populations of the species. These three red fox populations were most similar to each other morphologically and the closest geographically. By comparing the relationships in our phenogram with the phylogenetic tree of the relations between populations across the whole range of the red fox based on dental polymorphism [16], we can conclude that the western and southern European red foxes make up a distinct line from the central and east European populations. This observation suggests the Late Pleistocene red foxes of Belgium and recent Belgian and Italian populations share a common gene pool, and during the Last Glacial Maximum red foxes from western Europe survived in a southern–Italian refugium. The recent Polish population of red foxes was most probably populated by red foxes from some eastern refugium.

## Conclusions

Morphotype analysis of the lower carnassials in ancient and extant populations of the arctic and red fox revealed distinctive evolutionary trends in the two species. In the arctic fox the progressive changes in dental form tended towards hypercarnivory, whereas in the red fox the most primitive characters have been retained from the Pleistocene to recent times. Such evolutionary tendencies for both foxes are a result of their varying life strategies and evolutionary trajectories in the Late Pleistocene, and Holocene. The arctic fox occupies the most extreme habitats of the northern hemisphere, and over the last decades its range has been declining in response to pressure from the red fox.

## Supporting information

**S1 Table. List of the fossil first lower molars of the arctic fox (*Vulpes lagopus*) from RBINS, Brussels, used in the study with information about site excavation and left/right position of the molar in the tooth-row.**
(DOC)

**S2 Table. List of the fossil first lower molars of the red fox (*Vulpes lagopus*) from RBINS, Brussels, used in the study with information about site excavation and left/right position of the molar in the tooth-row.**
(DOC)

**S3 Table. AMS dates and their calibration (https://c14.arch.ox.ac.uk/oxcal/OxCal.html) of mammals present at the Belgian sites analyzed in this study (anthrop.: Anthropological).**
(DOC)

**S4 Table. List of dental morphotypes used to assess of dental polymorphism in the red fox (*Vulpes vulpes*) and the arctic fox (*Vulpes lagopus*).**
(DOC)

## Acknowledgments

The study was carried out in the Royal Belgian Institute of Natural Sciences in Brussels (BE-TAF) as a part of the SYNTHESYS Project (www.naturalsciences.be/coop/synthesys).

## Author Contributions

**Conceptualization:** Elwira Szuma.

**Formal analysis:** Elwira Szuma.

**Funding acquisition:** Elwira Szuma, Mietje Germonpré.

**Investigation:** Elwira Szuma.

**Methodology:** Elwira Szuma.

**Project administration:** Elwira Szuma.

**Resources:** Elwira Szuma, Mietje Germonpré.

**Visualization:** Elwira Szuma.

**Writing – original draft:** Elwira Szuma, Mietje Germonpré.

**Writing – review & editing:** Elwira Szuma.

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
