## [Decision Letter · Decision Letter 0]

29 Oct 2019

PONE-D-19-22849

Were ancient foxes far more carnivorous than recent ones? – carnassial morphological evidence

PLOS ONE

Dear Dr Szuma,

Thank you for submitting your manuscript to PLOS ONE. After careful consideration, we feel that it has merit but does not fully meet PLOS ONE’s publication criteria as it currently stands. Therefore, we invite you to submit a revised version of the manuscript that addresses the points raised during the review process.

Several changes, listed below, are requested by the reviewers. It should not include any supplementary studies of the dataset but improvements can be made on the presentation and discussion of your main results. Please consider those advices carefully and pay special attention to the presentation and discussion of the statistical findings supporting your conclusions.

We would appreciate receiving your revised manuscript by Dec 13 2019 11:59PM. To enhance the reproducibility of your results, we recommend that if applicable you deposit your laboratory protocols in protocols.io, where a protocol can be assigned its own identifier (DOI) such that it can be cited independently in the future. For instructions see: http://journals.plos.org/plosone/s/submission-guidelines#loc-laboratory-protocols

We look forward to receiving your revised manuscript.

Kind regards,

Cyril Charles

Academic Editor

PLOS ONE

Journal Requirements:

3. In your manuscript, please provide additional information regarding the specimens used in your study. Ensure that you have reported specimen numbers and complete repository information, including museum name and geographic location.

For more information on PLOS ONE's requirements for paleontology and archaeology research, see https://journals.plos.org/plosone/s/submission-guidelines#loc-paleontology-and-archaeology-research.

Reviewers' comments:

Reviewer's Responses to Questions

**Comments to the Author**

1. Is the manuscript technically sound, and do the data support the conclusions?

Reviewer #1: Yes

Reviewer #2: Yes

2. Has the statistical analysis been performed appropriately and rigorously? 

Reviewer #1: Yes

Reviewer #2: Yes

3. Have the authors made all data underlying the findings in their manuscript fully available?

Reviewer #1: Yes

Reviewer #2: Yes

4. Is the manuscript presented in an intelligible fashion and written in standard English?

Reviewer #1: Yes

Reviewer #2: Yes

5. Review Comments to the Author

Reviewer #1: 1) If possible, please list institutional catalog numbers for all specimens used in this study as supplementary data.

2) Lines 340, 342, and 388: Remove "n. sp." as this suggests that you name the species in this manuscript, which is not the case.

Reviewer #2: In this manusript, Szuma and co-author analyzes a large dataset of extant and fossil arctic and red fox carnassial specimens, using morphotype analysis and a carnivory index to understand morphological changes in fox populations during the Quaternary. This investigation thoroughly examines attributes of carnassial dentition size and morphology in fossil and extant specimens of two species (Vulpes lagopus and Vulpes vulpes) from multiple populations. The authors provide valuable data and intriguing inferences regarding ecological and evolutionary changes in these foxes. The analyses are based on a solid foundation of previous work by the first author and others, and provides an interesting perspective on dental evolution in an ecologically flexible lineage of carnivorans. I find the general topic and data to be worthy of publication, but there are issues with the presentation of statistical findings and organization of the discussion section that need additional work. Therefore, I am recommending major revisions to the manuscript before acceptance. Specific comments are included below:

Abstract:The title of the paper poses a specific question about evolutionary trends in the carnassial morphology of Vulpes, but neither the abstract nor the conclusions specifically answer this question. I suggest revision of both sections to more clearly present the implications of the findings from the analyses.

P3L39: recommend "specialized" rather than "special".

P3L41: the bite force / mechanical advantage is maximized in the most posterior teeth. It is more accurate to say that the carnassial tooth is in a position that simultaneously maximizes bite force AND gape.

P3L42: "is intimately associated with" rather than "accurately reflects", or include reference showing that carnassial actually accurately reflects diet.

Line 50 & Line 100: Please change “Up till” to “Until”

P5L107: "less carnivorous" would suffice, without using "specialised".

P5L108: "to present" rather than "to nowadays".

P6L118-119: Incorporating left and right teeth from the same individuals introduces statistical non-independence in the fossil samples. The authors should provide an explanation of how this method choice could affect study outcomes.

Line 176: Please replace “genders” with “sexes.” The word gender is often interpreted as meaning individuals’ concepts of themselves (in the human societal context), whereas the word sex typically refers to the biological differences of males and females.

Line 181-182: For improved clarity, I suggest writing “length of M1 (LM1) and width of M1 (WM1).”

P11L252: how could the p value be simultaneously smaller than 0.001 and larger than 0.05? Please clarify.

P12L279: A p>0.05 does not imply that the tested attributes "did not vary"; it simple means that samples with differences as large as those observed are expected by chance at higher percentages than the 0.05 level evaluated. Please clarify.

Lines 458-468: The concluding paragraph could be enhanced by adding 1-3 sentences summarizing the main findings regarding the arctic fox (Vulpes lagopus), given that the overall focus of the paper has the broader scope of ancient foxes.

Discussion section in general: lots of results are presented in the discussion section, but I would like to see more discussion of their implications. P18L385 specifically spells out the implications of the dental morphotype analyses in arctic foxes: "a progressive reduction in tooth size and an undermining or predatory specialization in the dental system"; I would like to see similarly clear statements about the red fox data. In addition, the discussion section lacks a summarizing paragraph to integrate the overall conclusions about the question posed in the title: were ancient foxes far more carnivorous than recent ones? Related to this point, the current manusript ends with a section on the "phylogeny of the arctic and red foxes", which is not the focus of this study (authors refer to other research, comparing them to their phenograms, but the data from this specific study does not resolve phylogenetic relationships because no phylogenetic analysis was conducted). Therefore, I recommend a reorganization of the discussion section to bring the focus of the paper back on track.

6. PLOS authors have the option to publish the peer review history of their article (what does this mean?). If published, this will include your full peer review and any attached files.

Reviewer #1: No

Reviewer #2: No

---

## [Author Response · Author response to Decision Letter 0]

3 Dec 2019

My answers to reviewer comments:

Reviewer #1: 

1) If possible, please list institutional catalog numbers for all specimens used in this study as supplementary data. fixed

2) Lines 340, 342, and 388: Remove "n. sp." as this suggests that you name the species in this manuscript, which is not the case. fixed

Reviewer #2: 

In this manusript, Szuma and co-author analyzes a large dataset of extant and fossil arctic and red fox carnassial specimens, using morphotype analysis and a carnivory index to understand morphological changes in fox populations during the Quaternary. This investigation thoroughly examines attributes of carnassial dentition size and morphology in fossil and extant specimens of two species (Vulpes lagopus and Vulpes vulpes) from multiple populations. The authors provide valuable data and intriguing inferences regarding ecological and evolutionary changes in these foxes. The analyses are based on a solid foundation of previous work by the first author and others, and provides an interesting perspective on dental evolution in an ecologically flexible lineage of carnivorans. I find the general topic and data to be worthy of publication, but there are issues with the presentation of statistical findings and organization of the discussion section that need additional work. Therefore, I am recommending major revisions to the manuscript before acceptance. Specific comments are included below:

Abstract:The title of the paper poses a specific question about evolutionary trends in the carnassial morphology of Vulpes, but neither the abstract nor the conclusions specifically answer this question. I suggest revision of both sections to more clearly present the implications of the findings from the analyses.

The manuscript was supplemented by conclusions about evolutionary trends in the carnassial morphology of Vulpes in discussion and separate last chapter "Conclusions" 

P3L39: recommend "specialized" rather than "special". fixed

P3L41: the bite force / mechanical advantage is maximized in the most posterior teeth. It is more accurate to say that the carnassial tooth is in a position that simultaneously maximizes bite force AND gape. fixed

P3L42: "is intimately associated with" rather than "accurately reflects", or include reference showing that carnassial actually accurately reflects diet. fixed

Line 50 & Line 100: Please change “Up till” to “Until”

 in these phrase 'up till' is much appropriate 

P5L107: "less carnivorous" would suffice, without using "specialised". fixed

P5L108: "to present" rather than "to nowadays". fixed

P6L118-119: Incorporating left and right teeth from the same individuals introduces statistical non-independence in the fossil samples. The authors should provide an explanation of how this method choice could affect study outcomes.

Each fossil first lower molar or part of mandible with the lower carnassial were identified as a separate specimen with own collection number (see Appendix 3, 4). It was impossible to these data found left and right molars of the same individual. Probably many of the teeth came from different individuals. Generally morphotype characters indicate left-right symmetry, also morphotypes groups P, R, S and K, L. However in some specimens we observed unilateral expression of morphotype characters. Determination of the influence of the method impact on the last study outcomes is very difficult in these circumstances. However the number of the ancient samples of the red fox (n=35) and the arctic fox (n=45) seems to be high enough to minimize the method affect. 

Line 176: Please replace “genders” with “sexes.” The word gender is often interpreted as meaning individuals’ concepts of themselves (in the human societal context), whereas the word sex typically refers to the biological differences of males and females. fixed

Line 181-182: For improved clarity, I suggest writing “length of M1 (LM1) and width of M1 (WM1)”. fixed

P11L252: how could the p value be simultaneously smaller than 0.001 and larger than 0.05? Please clarify. fixed

P12L279: A p>0.05 does not imply that the tested attributes "did not vary"; it simple means that samples with differences as large as those observed are expected by chance at higher percentages than the 0.05 level evaluated. Please clarify. fixed

Lines 458-468: The concluding paragraph could be enhanced by adding 1-3 sentences summarizing the main findings regarding the arctic fox (Vulpes lagopus), given that the overall focus of the paper has the broader scope of ancient foxes.

This part of discussion was extended and new citations were included.

Discussion section in general: lots of results are presented in the discussion section, but I would like to see more discussion of their implications. P18L385 specifically spells out the implications of the dental morphotype analyses in arctic foxes: "a progressive reduction in tooth size and an undermining or predatory specialization in the dental system"; I would like to see similarly clear statements about the red fox data. In addition, the discussion section lacks a summarizing paragraph to integrate the overall conclusions about the question posed in the title: were ancient foxes far more carnivorous than recent ones? Related to this point, the current manusript ends with a section on the "phylogeny of the arctic and red foxes", which is not the focus of this study (authors refer to other research, comparing them to their phenograms, but the data from this specific study does not resolve phylogenetic relationships because no phylogenetic analysis was conducted). Therefore, I recommend a reorganization of the discussion section to bring the focus of the paper back on track.

 fixed

With best regards,

Elwira Szuma

---

## [Editor Report · Decision Letter 1]

11 Dec 2019

Were ancient foxes far more carnivorous than recent ones? – carnassial morphological evidence

PONE-D-19-22849R1

Dear Dr. Szuma,

We are pleased to inform you that your manuscript has been judged scientifically suitable for publication and will be formally accepted for publication once it complies with all outstanding technical requirements.

With kind regards,

Cyril Charles

Academic Editor

PLOS ONE
---

## [Editor Report · Acceptance letter]

16 Dec 2019

PONE-D-19-22849R1 

Were ancient foxes far more carnivorous than recent ones? – carnassial morphological evidence 

Dear Dr. Szuma:

I am pleased to inform you that your manuscript has been deemed suitable for publication in PLOS ONE. Congratulations! Your manuscript is now with our production department. 

With kind regards,

on behalf of

Dr. Cyril Charles 

Academic Editor

PLOS ONE